# *Pseudopropionibacterium propionicum* as a Cause of Empyema; A Diagnosis with Next-Generation Sequencing

**DOI:** 10.3390/pathogens13020165

**Published:** 2024-02-12

**Authors:** Sumbal Babar, Emily Liu, Savreet Kaur, Juzar Hussain, Patrick J. Danaher, Gregory M. Anstead

**Affiliations:** 1Division of Infectious Diseases, Department of Medicine, Carilion Clinic, 2001 Crystal Spring Ave, Suite 301, Roanoke, VA 24014, USA; sbabar@carilionclinic.org; 2Long School of Medicine, University of Texas Health San Antonio, 7703 Floyd Curl Drive, San Antonio, TX 78229, USA; liue3@livemail.uthscsa.edu (E.L.); kaurs2@livemail.uthscsa.edu (S.K.); 3Internal Medicine Residency Program, University of Texas Health San Antonio, 7703 Floyd Curl Drive, San Antonio, TX 78229, USA; hussainj@uthscsa.edu; 4Division of Infectious Diseases, Department of Medicine, University of Texas Health San Antonio, 7703 Floyd Curl Drive, San Antonio, TX 78229, USA; patrick.danaher@va.gov; 5Division of Infectious Diseases, Medical Service, South Texas Veterans Health Care System, 7400 Merton Minter Blvd, San Antonio, TX 78229, USA

**Keywords:** *Pseudopropionibacterium propionicum*, empyema, culture negative, next-generation sequencing

## Abstract

*Pseudopropionibacterium propionicum* (*P.p*.) is an anaerobic, Gram-positive, branching beaded rod that is a component of the human microbiome. An infection of the thoracic cavity with *P.p.* can mimic tuberculosis (TB), nocardiosis, and malignancy. We present a case of a 77-year-old male who presented with dyspnea and a productive cough who was initially misdiagnosed with TB based on positive acid-fast staining of a pleural biopsy specimen and an elevated adenosine deaminase level of the pleural fluid. He was then diagnosed with nocardiosis based on the Gram stain of his pleural fluid that showed a Gram-positive beaded and branching rod. The pleural fluid specimen was culture-negative, but the diagnosis of thoracic *P.p*. infection was determined with next-generation sequencing (NGS). The patient was initially treated with imipenem and minocycline, then ceftriaxone and minocycline, and later changed to minocycline only. This report shows the utility of NGS in making a microbiological diagnosis when other techniques either failed to provide a result (culture) or gave misleading information (histopathologic exam, pleural fluid adenosine deaminase determination, and organism morphology on Gram stain).

## 1. Introduction

*Pseudopropionibacterium propionicum* (*P.p*.) is an anaerobic, Gram-positive, branching beaded rod that is a component of the human skin, oral, and gastrointestinal microbiome [1,2,3]. Infection with *P.p*. typically presents as a head and neck actinomycosis-like illness [2], but has also been implicated in other infections, including lacrimal canaliculitis [3,4], cervicofacial infections [5,6], tympanomastoiditis [7], pulmonary infections [8,9], osteomyelitis [10,11], a pelvic abscess related to an intra-uterine device [12], a psoas abscess [13], and a brain abscess [14,15]. An infection of the thoracic cavity with *P.p.* typically presents with dyspnea, weight loss, and night sweats, which can mimic tuberculosis (TB), nocardiosis, and malignancy [16]. The definitive identification of *P.p*. is complicated by its slow growth in a culture (5–20 days); although Gram stains are available sooner, its Gram-positive, branching beaded rod morphology leaves a broad differential diagnosis of various bacteria [3]. We present a case of a 77-year-old male who presented with dyspnea and a productive cough who was initially misdiagnosed with TB based on positive acid-fast staining of a pleural biopsy specimen and an elevated adenosine deaminase level of the pleural fluid. Then, he was diagnosed with nocardiosis based on the Gram stain of the pleural effusion. The pleural fluid specimen was culture-negative, but the diagnosis of thoracic *P.p*. infection was ultimately determined with next-generation sequencing (NGS). The purpose of this report is to show the utility of NGS in making a microbiological diagnosis when other techniques either failed to provide a result (culture) or gave misleading information (histopathologic exam, pleural fluid adenosine deaminase determination, and organism morphology).

## 2. Case Presentation

A 77-year-old male presented to an infectious diseases clinic with a productive cough of clear to white sputum for the past 1–2 months, dyspnea on exertion, a 27 kg weight loss, decreased appetite, and night sweats for the past six months. He had a past medical history of chronic obstructive pulmonary disease, chronic kidney disease, and anemia. He also had a history of a penicillin allergy from five years ago, which had presented as urticaria. He reported a 50-pack year history of smoking and had an extensive travel history including Central and South America, Asia, and Europe. He had multiple negative tuberculin skin tests in the past and denied any exposure to TB.

The patient had been living in West Virginia, where he had recently completed an extensive evaluation including a positron emission tomographic (PET) scan, which revealed a loculated left pleural effusion and adjacent pleural thickening. A biopsy specimen obtained from the pleura showed acute suppurative and granulomatous inflammation with possible abscess formation and rare acid-fast bacilli. After the PET scan, because of severe anemia, a bone marrow biopsy was performed, showing plasma cell dyscrasia.

His initial vital signs were a temperature of 36.1 °C, blood pressure of 115/59 mm Hg, pulse of 84, respiration rate of 23/min, and oxygen saturation of 97% on room air; his physical exam was unremarkable. Preliminary laboratory evaluation showed a white blood cell (WBC) count of 4.8 × 10^9^/L (reference range (RR): 4.0–10.0), hemoglobin level of 6.3 g/dL (RR: 13.5–17.5), platelet count of 148 × 10^9^/L (RR: 150–400), creatine level of 8.7 mg/dL (RR: 0.7–1.3), and potassium level of 7.0 mmol/L (3.5–5.1). He was admitted to the hospital and his initial CT scan of the chest showed nodular opacities with bilateral pleural effusions (Figure 1).

A PET scan showed a large, tracer avid loculated pleural effusion in the left lung base measuring 14.5 cm × 2.8 cm × 11.0 cm (Figure 2). A thoracentesis, followed by chest tube placement, drained 200 mL of purulent fluid (Figure 3). The adenosine deaminase (ADA) level of the empyema fluid was 714 U/L (RR < 9.2), consistent with TB. A renal biopsy was performed that showed acute interstitial nephritis and lambda light chain cast nephropathy. At the same time, the hematology service requested emergent clearance to start dexamethasone at 20 mg daily for the nephropathy due to plasma cell dyscrasia. Due to a concern about TB based on the prior histopathologic exam and the ADA result, the patient was started on empiric rifampin, isoniazid, pyrazinamide, and ethambutol (RIPE) therapy, pending further investigation.

The Gram stain of the pleural fluid showed beaded, branching Gram-positive rods, thought with visual inspection to be *Nocardia* spp. (Figure 4). However, the culture of the pleural fluid yielded no growth after 3 days, so a sample was sent for NGS (MicroGenDX, Orlando, FL, USA) to assist in definitively identifying a causative organism. The patient was started on intravenous ceftriaxone at 2 g daily and oral minocycline at 100 mg twice daily for presumed nocardiosis and the RIPE therapy was discontinued. Magnetic resonance imaging of the brain was negative for findings suggestive of nocardiosis. Meanwhile, 16S pyrosequencing (Quest Diagnostics, Secaucus, NJ, USA) was performed on paraffin-embedded tissue from the pleural biopsy specimen from the prior investigation conducted in West Virginia and this was negative for pathogens. The patient was discharged on regular hemodialysis sessions with IV imipenem and oral minocycline for presumptive pulmonary nocardiosis. A subsequent evaluation via the hematology service concluded that the patient did not meet the histopathologic and radiologic criteria for myeloma, and he was diagnosed with monoclonal gammopathy of renal significance.

After discharge, NGS performed on the empyema sample yielded a positive result for *Pseudopropionibacterium propionicum* (*P.p.*). The empyema fluid submitted for an AFB, fungal, and routine culture remained negative after 4 weeks of incubation. Two weeks after discharge, the patient was admitted to an outside hospital due to seizures on imipenem. Once stabilized, he was transferred back to our hospital; he had been off his antibiotics for one week as the other providers discontinued them due to not finding any active infection. A repeat CT showed worsening of his pulmonary situation, with an obstruction of the left mainstem bronchus and near complete collapse of the left lower lobe, bilateral pleural effusions, and peripheral atelectasis of the left upper lobe and lingula (Figure 5). The patient was started on ceftriaxone at 2 g and minocycline was continued. A bronchoscopy showed mucus plugging and a bronchoalveolar lavage did not identify *P.p.* on NGS, indicating the efficacy of the antimicrobials that had been received to date. Still, we elected to treat this infection like actinomycosis, with intravenous ceftriaxone for six weeks, followed by a planned six to eleven months of oral minocycline. He had completed nine months of treatment with an apparent resolution of the initial *P.p.* infection but died from sepsis from an intestinal perforation due to a strangulated hernia. The sequence of clinical events for this patient is summarized in Table 1.

## 3. Discussion

We report a case of a thoracic actinomycosis-like infection caused by *Pseudopropionibacterium propionicum* (*P.p.*). The patient’s extensive travel history on presentation and initial evaluation revealing an AFB-positive pleural biopsy specimen appeared consistent with a diagnosis of tuberculosis. A filamentous organism was observed in the Gram stain of the pleural fluid, which was thought to be *Nocardia* sp., but the pleural fluid culture was negative. However, NGS was able to identify the likely causative organism in the pleural fluid as *P.p.* in the absence of a positive culture. The early identification of *P.p.* as the causative organism could have facilitated timely antibiotic selection appropriate for the treatment of actinomycosis, potentially preventing our patient’s imipenem-induced seizures and subsequent readmission, and the premature discontinuation of antibiotics.

The species name of *Pseudopropionibacterium propionicum* has been changed several times since its original naming in 1896 as *Streptothrix israeli* [16,17,18,19]. It is important to be mindful of the organism’s previous names (Figure 6) because database searches using its current name yield few articles, necessitating searches using its older names. Prior names could also be a source of confusion: *Cutibacterium* spp. were previously referred to as cutaneous *Propionibacterium* spp., and thus clinicians may attribute the identification of *P.p*. in a specimen as a possible skin contaminant [16]. This is less of a concern now due to *P.p.*’s most recent name change but highlights the importance of proper research, identification, and knowledge of changing disease/pathogen nomenclature.

Many organisms stain acid fast so it is imperative to consider a wide differential diagnosis for AFB-positive organisms, especially if the AFB culture results are negative or pending. Other acid-fast or partial acid-fast organisms and structures include bacterial endospores, oocysts (from *Cryptosporidium*, *Cystoisospora*, and *Cyclospora*), *Rhodococcus equi*, *Corynebacterium* spp., *Nocardia* spp., *Legionella micdadei*, *Actinomyces israelii*, *Gordonia* spp., spores of the microsporidian *Nosema*, *Taenia saginata* eggs, hydatid hooklets, inclusion bodies in lead poisoning, and the heads of spermatozoa [20,21,22,23]. A retrospective analysis of positive AFB smears with subsequent negative cultures showed that while 62% were due to laboratory failure to isolate mycobacteria, up to 21% were false positives [24]. Thus, it is important to interpret AFB results within the context of the patient’s clinical characteristics.

The morphology of bacteria of the genus *Pseudopropionibacterium* is variable and not well known. It has been described as short diphtheroid rods, 0.2 to 0.8 µm in diameter by 1 to 5 µm long, with or without branching. The branching filaments may be 5 to 20 µm or more long. Swollen spherical cells, 5 to 20 µm in diameter, are formed by some strains. Microcolonies composed of long, branched septate and nonseptate filaments may also be observed [17].

This case also illustrates the difficulty of identifying *P.p*. with traditional staining and culture methods. We initially sent the patient’s pleural biopsy samples with rare acid-fast bacilli for 16s RNA pyrosequencing, which allows for a rapid identification of AFB isolates with 94–98% accuracy for *Mycobacterium* and *Nocardia* isolates [25]. The Quest Diagnostics pyrosequencing technique is based on the amplification of specific 16S rRNA sequences using AFB PCR primers; short sequences of 30–50 bases are identified and compared to databases to determine the AFB genus and species. Pyrosequencing is considered to be highly accurate and sensitive, as few as 100 AFB are able to produce a positive sequencing result [26]. However, in this case, the technique failed to establish a diagnosis because it lacked the primers to identify *P.p.* A second potential problem is using paraffinized tissue for the pyrosequencing. Formalin-fixation and paraffin embedding affects the quality of DNA due to cross-linking and heat damage, which may hinder the polymerase chain reaction [27]. For AFB-smear-negative specimens and weakly acid-fast staining organisms (e.g., *Nocardia* sp.), the GenXpert assay or Hsp65 deoxyribonucleic acid sequencing has been used for *Mycobacterium tuberculosis* detection and bacterial speciation, respectively [28,29].

We were ultimately able to identify *P.p.* as the causative organism in this case with NGS, demonstrating its utility in the diagnosis of difficult-to-culture organisms. There are multiple reasons why an infected clinical specimen may produce a negative culture result. Anaerobic bacteria may require special collection and culturing techniques. Some bacteria are fastidious and have specific nutritional requirements. Receiving antibiotics prior to sample collection may also result in negative cultures. In some cases, bacteria may exist in a viable but non-culturable state. Specimens with bacteria sequestered within a biofilm may also fail to grow in a culture [30].

To circumvent the problem of culture negativity, the technique of NGS has been developed. In this technology, thousands to billions of DNA fragments are simultaneously sequenced to determine the identity of microbes in a specimen. Since its advent in 2004, the cost of NGS has decreased by several orders of magnitude, now making it practical and cost-effective when other routine techniques fail to provide a diagnosis [31]. The specimen in this case was sent to MicroGenDX, which uses the Illumina MiSeq NGS platform; results are referenced against a validated database of >57,000 microbes [32]. As compared to PCR-based methods, NGS provides unbiased pathogen detection via direct sequencing of the specimen’s extracted DNA; this can facilitate the identification of rare or uncultured pathogens [33].

*Pseudopropionibacterium propionicum* is uniformly susceptible to penicillins, minocycline, and clindamycin in vitro [34]. Because this organism behaves like the more common actinomycetes, treatment guidelines for actinomycosis are followed. Generally, mild actinomycosis is treated with initial oral therapy, but severe or extensive disease, such as thoracic actinomycosis, requires parenteral therapy for 2–8 weeks followed by oral therapy for 6–12 months, with or without surgical source control [35,36]. The preferred regimen for severe actinomycosis involves high-dose penicillin, generally 18–24 million units daily for 2–6 weeks, followed by oral therapy with penicillin V or amoxicillin for 6–12 months. Alternatives include ceftriaxone or amoxicillin; those with beta-lactam allergies can be treated with tetracyclines, clindamycin, or imipenem [35].

## Figures and Tables

**Figure 1 pathogens-13-00165-f001:**
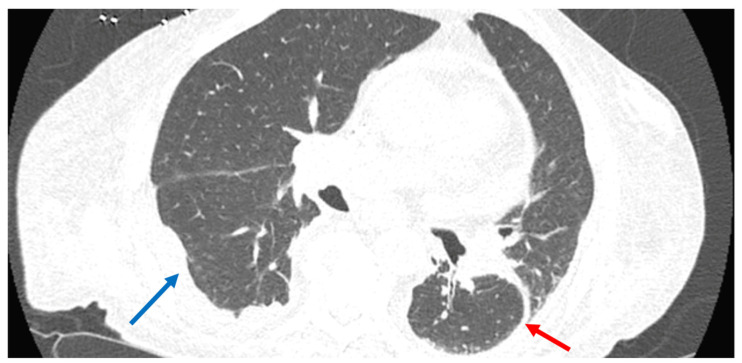
CT scan of the chest on admission showing small right pleural effusion (blue arrow) and a loculated left pleural effusion with associated pleural and left hemidiaphragm thickening (red arrow). There was also an infiltrate in the lingula.

**Figure 2 pathogens-13-00165-f002:**
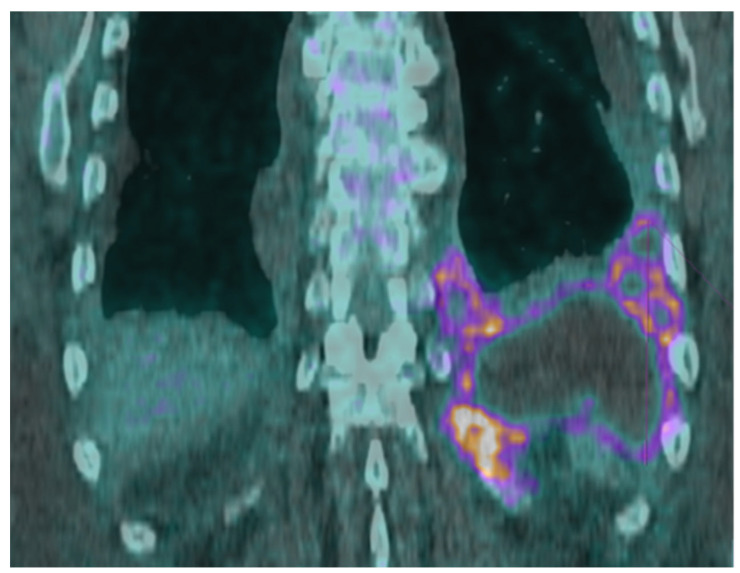
PET scan performed 4 days after the initial admission showing a large, tracer avid loculated pleural effusion in the left lung base measuring 14.5 cm × 2.8 cm × 11.0 cm, most consistent with chronic inflammation or infection. There was no primary lung mass or frank bony destruction, though there was some localized bone remodeling/periostitis.

**Figure 3 pathogens-13-00165-f003:**
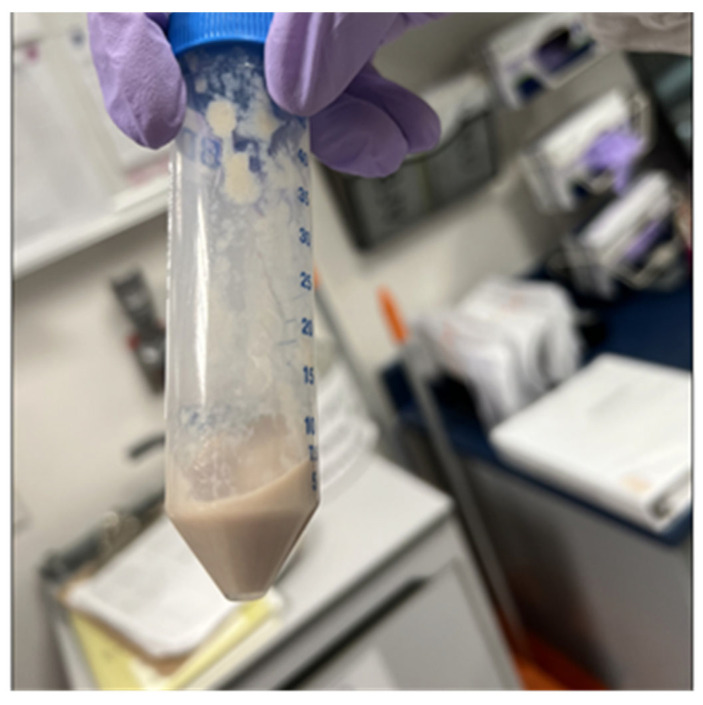
Purulent fluid obtained from the initial thoracentesis.

**Figure 4 pathogens-13-00165-f004:**
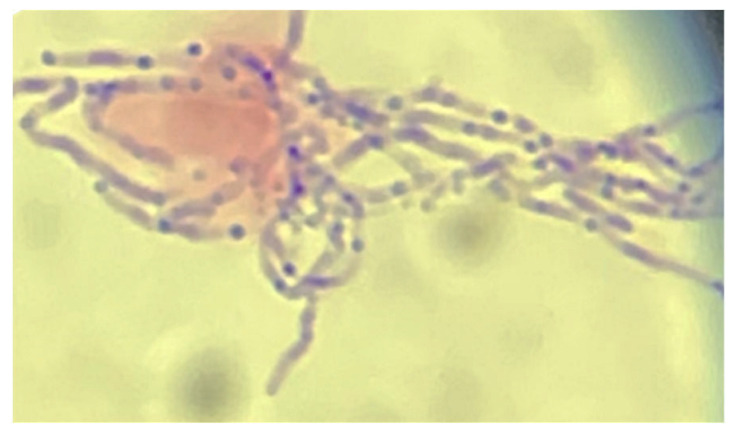
Photomicrograph of the Gram stain of the pleural fluid showing beaded, branching Gram-positive rods initially thought to be *Nocardia* sp.

**Figure 5 pathogens-13-00165-f005:**
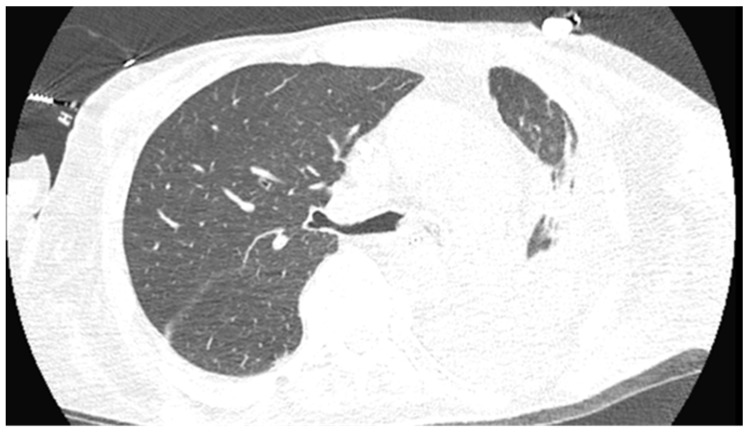
Repeat CT scan showed obstruction of the left mainstem bronchus and near complete collapse of the left lower lobe, bilateral pleural effusions, and peripheral atelectasis of the left upper lobe and lingula.

**Figure 6 pathogens-13-00165-f006:**
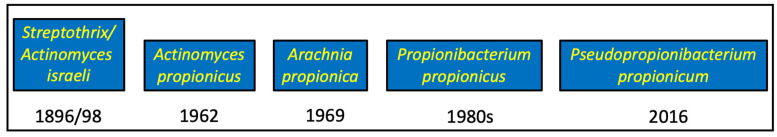
Taxonomic/nomenclature changes in *Pseudopropionibacterium propionicum* through the years.

**Table 1 pathogens-13-00165-t001:** Timeline of Clinical Events for the Case Patient.

Day	Clinical Events
−59	- Presents with chronic cough, weight loss, dyspnea at outside hospital - PET scan shows pleural effusion and pleural thickening
−39	- Pleural biopsy shows necrotizing granulomas and rare acid-fast bacilli (AFB) - For severe anemia, bone marrow biopsy was performed, showing plasma cell dyscrasia
1	- Admitted to our hospital for elevated creatinine (8.7 mg/dL)
2	- Hemodialysis started - CT scan shows loculated left pleural effusion with pleural and left hemidiaphragm thickening - Paraffin block from previous pleural biopsy sent for AFB pyrosequencing
4	- PET scan shows a large, tracer avid loculated pleural effusion in the left lung base
8	- Pleural effusion tapped; chest tube placed for empyema; pleural fluid shows high ADA - Empyema fluid sent for Gm stain and culture and AFB stain and culture
11	- Renal biopsy shows acute interstitial nephritis, lambda light chain cast nephropathy - Started dexamethasone for nephropathy - Started rifampin–isoniazid–pyrazinamide–ethambutol (RIPE) for presumptive TB
12	- Culture of empyema fluid is negative; Gram stain shows beaded Gram-positive rods, - suspected to be *Nocardia* sp. - Sample of empyema fluid sent for next-generation sequencing (NGS) - AFB pyrosequencing was negative; stopped RIPE
19	- Discharged from hospital on hemodialysis - Discharged on imipenem and minocycline for presumed pulmonary nocardiosis
24	- NGS Report: *Pseudopropionibacterium propionicum* (*P.p.)*
36	- Negative 4-week cultures for AFB, fungi
38	- Seizure on imipenem; off antibiotics for 1 week
46	- CT showing left mainstem bronchus obstruction, collapse of left lower lobe, bilateral pleural effusions, and peripheral atelectasis of the left upper lobe and lingula - Bronchoscopy performed and cleared mucus plugging; BAL was negative for *P.p.* with NGS
47	- Decision to treat *P.p.* lung infection like actinomycosis - Started ceftriaxone and minocycline for 6 weeks and then continued on minocycline
279	- Strangulated hernia, sepsis, and death

## Data Availability

The data are contained within the article.

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
