# Peer review of "Pseudopropionibacterium propionicum as a Cause of Empyema; A Diagnosis with Next-Generation Sequencing"

_pathogens, 2024, doi:10.3390/pathogens13020165_

Round 1
Reviewer 1 Report
Comments and Suggestions for Authors
This is a case report of Pseudopropionibacterium propoonicum causing lung empyema in a patient newly diagnosed with multiple myeloma. There was a time gap to make a correct diagnosis due to negative culture or other tests which may have caused side effects to the patient. The report is very well written and includes important teaching points. I have only a few minor comments which could make this report better.
1. Case report – they may consider adding a sentence about treatment of multiple myeloma in addition to dexamethasone.
2. Case report – line 91 – “, and a chest tube was placed.” – it is a repetition and can be removed.
3. Discussion – lines 170-174. The authors can consider adding a sentence to discuss how a paraffin-embedded tissue can affect the sensitivity of PCR.
4. If space allows, may consider adding a figure to illustrate the timeline of the clinical course.
Author Response
- Case report – they may consider adding a sentence about treatment of multiple myeloma in addition to dexamethasone.
(Edits highlighted in yellow on revised manuscript.)
We have provided additional details of his hematologic diagnosis. He was ultimately found not to meet the criteria of myeloma and was diagnosed with monoclonal gammopathy of renal significance. This is now indicated in the text. This newly introduced information does not change any analysis or conclusions about the infectious diseases aspects of the case, but provides clarity about the patient’s overall medical condition.
- Case report – line 91 – “, and a chest tube was placed.” – it is a repetition and can be removed.
The second statement on the chest tube placement was removed.
- Discussion – lines 170-174. The authors can consider adding a sentence to discuss how a paraffin-embedded tissue can affect the sensitivity of PCR.
SENTENCE ADDED: A second potential problem in this case is using paraffinized tissue for the pyrosequencing. It is known that formalin-fixation/paraffin embedding can affect the quality of DNA due to cross-linking and heat damage, which may hinder the polymerase chain reaction.
- If space allows, may consider adding a figure to illustrate the timeline of the clinical course
A table has been added to summarize the chronology of the major clinical events for this patient.
Reviewer 2 Report
Comments and Suggestions for Authors
The description concerns a very interesting case of infection of the pleural cavity by Pseudopropionibacterium propionicum bacteria. The authors diagnosed the cause using the Next Generation Sequencing (NGS) method.
The manuscript should be taken into consideration for publishing, but only after taking into account the following:
1. Did the NGS indicate only one species of bacteria (that would be quite surprising). If not, what percentage of readings was P. propionicum obtained? How to explain the presence of other species (if any)?
2. The term 'flora' is a misnomer because it refers to plants. Please replace with 'microbiota'
3. The name Gram must be capitalized, as it comes from the name of the creator of the bacterial staining method
Author Response
- Did the NGS indicate only one species of bacteria (that would be quite surprising). If not, what percentage of readings was P. propionicum obtained? How to explain the presence of other species (if any)?
Yes, the NGS did indicate Pseudopropioniobacterium propionicum as a single species.
- The term 'flora' is a misnomer because it refers to plants. Please replace with 'microbiota'
Flora has been replaced with microbiome.
- The name Gram must be capitalized, as it comes from the name of the creator of the bacterial staining method
For the “Gram stain” and gram-positive, we adopted the CDC preferred usage, which states:
“Gram should be capitalized and never hyphenated when used as Gram stain; gram negative and gram positive should be lowercase and only hyphenated when used as a unit modifier.”
Available online: https://wwwnc.cdc.gov/eid/page/preferred-usage#:~:text=Gram%20should%20be%20capitalized%20and,used%20as%20a%20unit%20modifier. (Accessed 25 Jan 2024).
In this paper, gram-positive is used as a unit modifier for bacteria.